# Testing the Tenacity of Small Ruminant Lentiviruses In Vitro to Assess the Potential Risk of Indirect Fomites’ Transmission

**DOI:** 10.3390/v17030419

**Published:** 2025-03-14

**Authors:** Maksym Samoilenko, Vitalii Nedosekov, Giuseppe Bertoni

**Affiliations:** 1Institute of Virology and Immunology IVI, 3147 Mittelhäusern, Switzerland; maksym.samoilenko@ivi.admin.ch; 2Institute of Virology and Immunology IVI, 3012 Bern, Switzerland; 3Vetsuisse Faculty, Department of Infectious Diseases and Pathobiology, University of Bern, 3012 Bern, Switzerland; 4Department of Epizootiology, Microbiology and Virology, Faculty of Veterinary Medicine, National University of Life and Environmental Sciences of Ukraine, 03041 Kyiv, Ukraine; nedosekov06@gmail.com; 5Graduate School for Cellular and Biomedical Sciences, University of Bern, 3012 Bern, Switzerland; 6Royal Veterinary College, University of London, London NW1 0TU, UK

**Keywords:** small ruminant lentiviruses, CAEV, MVV, VMV, tenacity, stability, replication capacity, transmission, fomites

## Abstract

In 2011–2013, we isolated and characterized small ruminant lentiviruses (SRLVs) from two flocks, one of goats and the other of sheep, that had never been in direct contact. Phylogenetic analysis of these viruses indicated a common origin, which led us to hypothesize indirect transmission of these viruses between the two flocks. Since, to our knowledge, there are no published data on the tenacity of these viruses, we started this work. In the first part, we monitored the loss of infectivity of two prototypic SRLV strains, MVV 1514 and CAEV-CO, over time, in liquid suspension. As expected, the suspensions stored at 4 °C better preserved the infectivity of the viruses. Additionally, viruses resuspended in milk, the medium mirroring the in vivo situation, proved more tenacious than those maintained in a cell culture medium. These viruses, subjected to harsh treatments such as drying and resuspending, partially maintained their replication capacity. After an immediate loss of nearly 1 log_10_ TCID_50_ immediately after desiccation, the viruses maintained their replication capacity for at least three weeks when desiccated in milk. These results suggest that fomites, clothing, or pastures contaminated with secretions or milk from infected animals might mediate the infection of animals independently of direct contact.

## 1. Introduction

Maedi-visna virus (MVV) and caprine arthritis-encephalitis virus (CAEV) belong to the genus Lentivirus of the *Retroviridae* family. Due to their seemingly different hosts and discovery times, these viruses were long thought to be species-specific to sheep and goats, respectively. Further investigation has revealed that most small ruminant lentivirus (SRLV) genotypes are promiscuous and are now assigned to the group of SRLV. SRLVs are geographically widely distributed and consist of the following five different genotypes: A (prototypical MVV), B (prototypical CAE), and the more recently discovered C, D, and E genotypes, with each genotype further divided into subtypes [1,2,3,4,5,6,7,8,9,10,11,12,13]. Trading is considered the principal factor in the worldwide dissemination of these viruses, the most prominent example being the importation of infected sheep to Iceland in the year 1933, which led to the infection of the local sheep and the first description of the pathological signs induced by SRLVs such as Maedi (dyspnea) and Visna (wasting) [14,15]. Recently, however, a study failed to detect the Icelandic subtype (A1) in German sheep, questioning the introduction of the virus into Iceland through the importation of Karakul sheep from Germany [16]. This notwithstanding, MVV caused substantial economic losses in Iceland, being responsible for the death of approximately 100,000 sheep [17,18].

The pathogenesis of these viruses is similar and influenced by factors such as the virus’s genotype and subtype, the infected host’s genetics, and the management of the flock. Except for rare acute manifestations, such as myeloencephalitis in young kids, SRLVs have common pathogenic traits, such as a long clinical latency of months to years, a subtle development of pathologic lesions, and a slow clinical evolution. The organs affected are similar in the two species, but the lesions and, thus, the clinical manifestations are distributed differently. Interstitial pneumonia with progressive respiratory failure and wasting predominate in sheep, while multifocal arthritis, involving, in particular, the carpal joint, prevails in goats [19,20,21]. The mammary gland is affected in both species, with interstitial, indurative mastitis being the cause of important economic losses and epidemiologic consequences by reducing the quantity and quality of milk and mediating an efficient vertical transmission of the infection [22,23,24]. The abundance of macrophages in colostrum and the pronounced tropism of SRLVs for these cells ensure the efficient infection of kids via colostrum intake [25]. Horizontal transmission is a proven way of transmission, especially in young animals, and plays a prominent role in sheep, where vertical transmission may be less important [26,27].

Several routes of virus transmission are described in the literature, which, in addition to the colostral and respiratory routes, also contemplates venereal transmission, embryo transfer, iatrogenic transmission, and, more controversially, intrauterine transmission [25,28]. Transmission via fomites is described, but its importance is considered minor, although difficult to assess, mainly because of the lack of information on the tenacity of these viruses when exposed to environmental factors [29].

In 2011–2013, we analyzed a case of virus transmission from a flock of sheep to goats in an epidemiological context that tended to rule out any direct contact, suggesting possible transmission of infection via fomites [30]. To confirm or refute the plausibility of this hypothesis, in this work, we set out to characterize the tenacity of the virus when exposed to conditions similar to those potentially present in the field. Milk was used as the vehicle for the virus because lactogenic infection is considered one of the most efficient mediators of infection, and aerosols generated during milking could also transmit the infection [31]. SRLVs can be isolated from cell-free milk and milk cells, and the viral load in milk-associated cells is much higher than that observed in other tissues [32,33]. In analogy to experiments performed with HIV, where the virus showed a surprising sturdiness in dry blood [34,35], we tested the tenacity of SRLVs in wet and dry conditions using milk or tissue culture medium spiked with the virus.

## 2. Materials and Methods

### 2.1. Virus

Caprine arthritis-encephalitis virus (CAEV-CO), accession number M33677 (CEAVCG), was selected for our experiments. This virus is the prototypical representative of SRLV subtype B1. The virus was generated by transfecting goat synovial membrane (GSM) cells with a molecular clone of the virus with restored dUTPase activity, generously provided by Yahia Chebloune [36,37,38]. A viral stock of 6.125 log_10_ TCID_50_ was used for all of the experiments. Maedi-visna virus (MVV 1514: Visna virus Icelandic strain 1514), accession number M60609 [39], was a generous gift from Prof. Sergio Rosati, Università degli Studi di Torino. This virus belongs to the A1 subtype of SRLVs. It was first isolated from a sheep and is the prototypical virus causing maedi-visna in sheep. A viral stock of 5.8 log_10_ TCID_50_ was used for all of the experiments.

### 2.2. Nutrient Medium

Dulbecco’s Modified Eagle Medium (DMEM—Seraglob, Switzerland, lot 410/631450), supplemented with 10% fetal bovine serum (FBS) (PAAN Biotech, Aidenbach, Germany, lot 190902), 1% Non-Essential Amino Acid (Seraglob, Switzerland, lot 410/821695), 0.5% of Penicillin–Streptomycin (Seraglob, Switzerland, lot 610/134072), and 1% of Amphotericin B (Seraglob, Switzerland, lot 610/763549) was used.

### 2.3. Drying Medium

As described above, DMEM supplemented with 10% FBS was used as a standard drying medium. To mimic natural conditions, commercially available pasteurized goat milk (milk), with a fat content of 3.5% (Molkerei Biedermann AG, Bischofszell, Switzerland), was also used.

### 2.4. Cells

GSM cells were used for the cultivation and titration of CAEV, while primary lamb synovial membrane (LSM) cells were used for the cultivation and titration of MVV. GSM and LSM cells were obtained as described previously [40].

### 2.5. Virus Stored in Liquid

Nutrient medium and milk were supplemented with 2.5% Penicillin–Streptomycin (Seraglob, Switzerland, lot 610/134072) and 1% of Amphotericin B (Seraglob, Switzerland, lot 610/763549) to prevent possible bacterial or fungal contamination. A working dilution of viruses was prepared by resuspending viral stocks in a nutrient medium or milk. The dilutions were prepared in a volume of 3 mL for every condition and every foreseeable timepoint and then aliquoted (1 mL into 1.5 mL Eppendorf tubes) and stored under the following conditions:

Room temperature (RT)—tubes were stored at ambient room temperature (18–25 °C) and were shielded from direct sunlight. At 4 °C—tubes were stored in a refrigerator in the dark at a stable temperature of 4 °C. At the control time points (7, 14, and 21 days), three tubes for each condition were taken and titrated by limiting the endpoint dilution. In our Institute’s accreditation frame, the temperatures in the laboratories and refrigerators where the viruses were stored were monitored daily. To assess the tenacity of the viruses, 3 tubes for both milk and the nutrient medium and at each temperature were taken and titrated via 10-fold dilution in quadruplicates on 96-well tissue culture plates that were previously seeded with primary cells (LSM for MVV and GSM for CAEV) at 50% confluency.

The plates were incubated for 12 days and stained using the Hemacolor^®^ staining kit (Hemacolor^®^, Merck, Germany), according to the manufacturer’s instructions. At this point, the plates were scored under a brightfield microscope for the presence of syncytia. The Spearman–Kärber method was used to calculate the titers [41].

### 2.6. Virus Dried, Stored, and Resuspended

The virus stock dilutions were prepared as outlined in the preceding section, and the drying process was carried out in standard cell-culture 6-well plates (TPP AG, Trasadingen, Switzerland). The virus-containing milk or medium was inoculated into the empty wells of the 6-well plate, which was then placed into a biosafety laminar-flow cabinet with an open lid and left to dry at room temperature for 8 h. The day on which the plates were prepared and dried was designated as day 0.

The plates containing the dried viruses were then stored at room temperature (18–25 °C) or in a refrigerator at 4 °C, shielded from direct sunlight. At each control point (0, 7, 14, and 21 days), three plates from both RT and 4 °C for each virus and the two drying media were tested for replication-competent viruses.

The dried residue in each well was resuspended by adding 1 mL of DMEM+FBS, and 100 µL of the resuspended material was transferred into an Eppendorf tube containing 900 µL of fresh cell culture medium. A 10-fold dilution of the material was performed and finally transferred in quadruplicates to 96-well tissue culture plates seeded with primary cells (LSM for MVV and GSM for CAEV) at 50% confluency. The plates were incubated at 37 °C in a 5% CO_2_ atmosphere for 8 h, after which the supernatant was substituted with 100 μL of fresh DMEM + FBS medium.

Incubation, staining, and readout were performed as outlined in Section 2.6.

### 2.7. Statistical Analysis

All statistical analyses and visualizations were performed using GraphPad Prism version 10.1.2 for Windows (GraphPad Software, San Diego, CA, USA). The Shapiro–Wilk test was used to assess the normality of the data distribution. Two-way ANOVA was performed to analyze group differences in the time-course experiments. A Student’s *t*-test was used to evaluate the immediate effect of drying on virus infectivity. A *p*-value of less than 0.05 (*p* < 0.05) was considered statistically significant for all analyses.

## 3. Results

### 3.1. Virus Stored in Liquid

As shown in Figure 1a for MVV and Figure 1b for CAEV, the viruses incubated in the medium and milk at 4 °C were still infectious after three weeks, losing about 1 log_10_ TCID_50_. In contrast, storage at room temperature dramatically impacted infectivity. MVV lost its replication capacity after 2 weeks in the medium and about 2 log_10_ TCID_50_ in the milk. After three weeks, no infectious MVV could be recovered. CAEV was no longer infectious after 3 weeks of incubation at RT in the medium, but infectious viruses could be recovered when incubated in milk.

### 3.2. Virus: Dried, Stored, and Resuspended

The drying process impacted the infectivity of both viruses, which lost about half to 1 log_10_ TCID_50_ immediately after drying. As for the viruses in liquid suspension, the storage temperature substantially impacted the tenacity of these viruses, as shown in Figure 2a for MVV and Figure 2b for CAEV. For both viruses, the presence of milk during the drying process showed a protective effect, permitting the recovery of infectious viruses for a more extended time period. This substantial protective role of milk is illustrated in Figure 3a for MVV and Figure 3b for CAEV.

## 4. Discussion

In contrast to other viruses, to our knowledge, the tenacity of small ruminant lentiviruses has not yet been studied in detail [42]. The ability of these viruses to maintain their replication capacity (a term adopted from reference [42]) in the environment when exposed to a hostile milieu determines their potential ability to infect animals without direct contact, such as by grazing on a pasture previously used by infected animals or through contamination of caretakers’ clothing or fomites.

For our experiments, we chose two prototypic viruses for the two main genotypes, A (MVV 1514) and B (CAEV-CO), aware that this may limit the generalization of our results. Primary isolates would have been ideal, but these viruses often have a strict tropism for macrophages, making their titration extremely cumbersome.

Our experiments demonstrate the relative resilience of these viruses even under harsh conditions, such as drying and storage for several days before reconstitution and titration. For all conditions, milk appears to stabilize the viruses, and, not surprisingly, the lower storage temperature also positively affects the preservation of infectivity. This is in line with previously published observations of HIV, showing that the infectivity of the virus was preserved for a more extended period in blood or plasma compared to tissue culture medium [34].

The milk of SRLV-infected animals contains high titers of virus and numerous virus-infected cells, such as macrophages, that may contaminate the environment, pastures, fodder, and hands and clothes of people attending sheep and goats [32]. We also attempted to reproduce these results using infected cells instead of infectious virus in suspension.

The variability between the samples was too large to obtain significant results, but a clear trend mirrored the results obtained with viruses in suspension. The tenacity of these viruses, demonstrated in this work, makes their transmission from flock to flock by fomites or shared pastures plausible. This validates our original hypothesis that genetically very closely related viruses detected in two strictly separated flocks of sheep and goats may be a case of fomite-mediated transmission [30,43]. The application of strict biosafety rules will prevent such occurrences in the future. Similarly, circulating SRLVs in wild ruminants may originate from pasture contamination rather than direct contact between wild and farmed animals [44].

In conclusion, we must emphasize that our results do not yet allow us to determine the actual risk of SRLV transmission via fomites. Indeed, to answer this question, it will be necessary to establish the minimal infectious titer of these viruses in vivo, especially by the oral route. Indeed, the infectious titers required to infect an animal were shown to be very low for intratracheal infection but were estimated to be much higher for intranasal infection and the oral route [31,45].

## Figures and Tables

**Figure 1 viruses-17-00419-f001:**
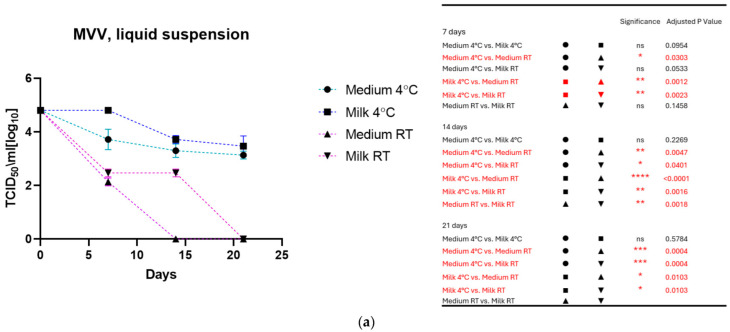
(**a**) (MVV) and (**b**) (CAEV). The loss of infectivity of the virus suspensions over time is illustrated. Regardless of the storage liquid, the temperature of 4 °C has a noticeable protective effect. Remarkably, milk preserves the infectivity of virus suspensions better than the culture medium at room temperature. This was particularly evident for CAEV in Figure 1b. The table on the right shows the pairings with significant differences and the respective *p*-values, calculated by two-way ANOVA (ns: not significant, *—*p*-value < 0.05, **—*p*-value < 0.01, ***—*p*-value < 0.001, ****—*p*-value < 0.0001).

**Figure 2 viruses-17-00419-f002:**
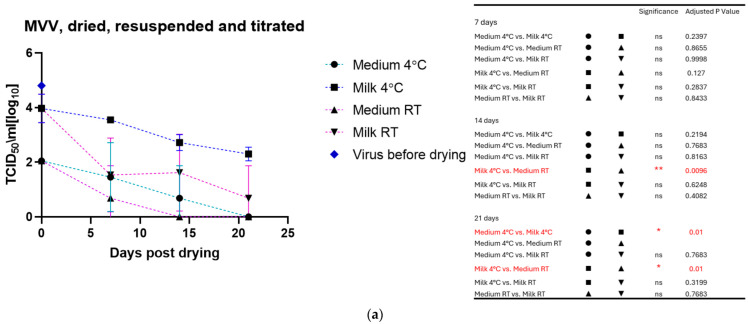
(**a**) (MVV) and (**b**) (CAEV). Virus suspensions were dried, stored for different periods, and resuspended immediately before titration. The blue square shows the titer of the virus stock used. Storage in dry conditions induces a loss of infectivity over time, which is more pronounced at RT than at 4 °C. Independently of the storage temperature, milk shows an evident protective role in the infectivity of the stored virus preparations. The table on the right shows the pairings with significant differences and the respective *p*-values, calculated by two-way ANOVA (ns: not significant, *—*p*-value < 0.05, **—*p*-value < 0.01, ****—*p*-value < 0.0001).

**Figure 3 viruses-17-00419-f003:**
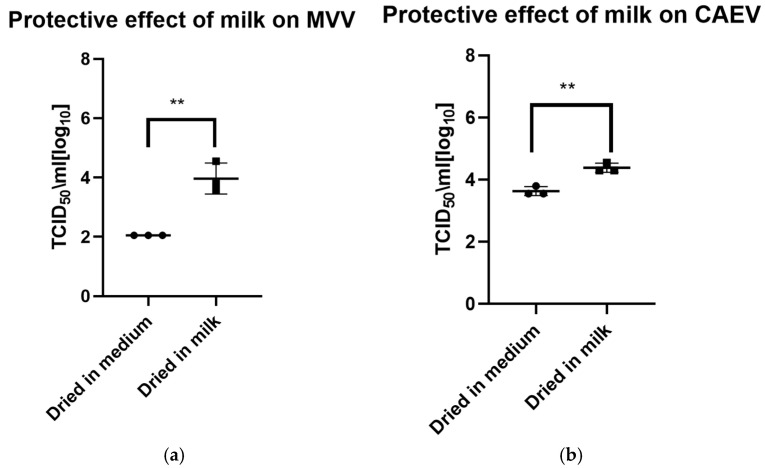
(**a**) (MVV) and (**b**) (CAEV). Virus suspensions in tissue culture medium or milk were dried for 8 h. The following day, the dried material was rehydrated and resuspended in the same volume as before drying and immediately titrated. Milk showed a robust protective effect on the infectivity of MVV and CAEV, with a *p*-value of 0.0031 ** in both cases.

## Data Availability

Raw data are available upon request at: maksym.samoilenko@unibe.ch.

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
