# Peer review of "Testing the Tenacity of Small Ruminant Lentiviruses In Vitro to Assess the Potential Risk of Indirect Fomites’ Transmission"

_viruses, 2025, doi:10.3390/v17030419_

Round 1
Reviewer 1 Report
Comments and Suggestions for Authors
The authors present a well written paper with clear results showing that infectivity of SRLV is stable in milk especially in cool conditions (4 °C) and that this is maintained when the suspension is dried. This helps to explain transmission of virus between animals that have had no contact but may have shared areas/pasture at different times.
The paper is entitled ‘On the tenacity of small ruminant lentiviruses’. I think that in english tenacity is probably the wrong adjective and that this would be better expressed as stability or persistence.
There were a few of typographical errors that should be corrected.
Line 37 – say four genotypes but then name five
Line 50 – kits should be kids
Line 199-200 – need to rephrase – if there is too much variability to show the results are not significantly different cannot say it mirrors the included figures. Something like the following would be better ‘The results did not show significant differences but there was a trend that mirrored that shown by virus in suspension.’
Author Response
Reviewer #1
Comments and Suggestions for Authors
The authors present a well written paper with clear results showing that infectivity of SRLV is stable in milk especially in cool conditions (4 °C) and that this is maintained when the suspension is dried. This helps to explain transmission of virus between animals that have had no contact but may have shared areas/pasture at different times.
Response 1: Thank you very much for your work, which helped us improve our manuscript.
The paper is entitled 'On the tenacity of small ruminant lentiviruses'. I think that in english tenacity is probably the wrong adjective and that this would be better expressed as stability or persistence.
Response 2: We appreciated this comment because this is a topic that we discussed intensively while writing the manuscript. We modified the title as suggested by one of your colleagues, but we kept the term "tenacity" for the following reasons:
We have noticed that in the literature, "tenacity" is used predominantly by German-speaking or French-speaking authors. At the same time, authors of other mother tongues also use this term, but it is interchangeably used with "persistence" or "stability." In our opinion, the term persistence is not ideal as it creates unnecessary ambiguities with its use in virology for viruses that persist in the host using different strategies to evade the immune response, either by inducing immunotolerance like pestiviruses or by taking refuge in latency like herpes. In the following reference by two American authors, we found a definition that perfectly fits our concept of tenacity:
D.E. Stallknecht & J.D. Brown, Rev. sci. tech. Off. int. Epiz., 2009, 28 (1), 59-67: "Although the transmission of avian influenza (AI) viruses within both wild and domestic avian populations can be linked to environmental sources, information on their tenacity, or the ability of these viruses to remain infective outside of the host, is limited."
Stability would be acceptable, but we still prefer "tenacity" if the reviewer allows it.
There were a few of typographical errors that should be corrected.
Response 3: Thank you for finding these "persistent" slips that always manage to escape our attention.
Line 37 – say four genotypes but then name five
Response 4: corrected on page 1, line 39.
Line 50 – kits should be kids
Response 5: corrected on page 2, lines 52 and 63.
Line 199-200 – need to rephrase – if there is too much variability to show the results are not significantly different cannot say it mirrors the included figures. Something like the following would be better 'The results did not show significant differences but there was a trend that mirrored that shown by virus in suspension.'
Response 6: We agree with the reviewer and modified the sentence as follows (lines 219-220):
The variability between samples was too large to obtain significant results, but a clear trend mirrored the results obtained with viruses in suspension (data not shown).
Reviewer 2 Report
Comments and Suggestions for Authors
MAJOR COMMENTS
The article examines the persistence of small ruminant lentiviruses (SRLVs), in particular Visna Virus (MVV) and caprine encephalitis virus (CAEV). The researchers analysed the duration of infectivity of these viruses under different conditions and found that they persisted longer in milk suspensions than in cell culture media. Even after drying, the viruses retained their ability to replicate for weeks. This suggests that indirect transmission via contaminated fomites or pastures could occur without direct contact between animals. The study covers a gap in data regarding the survival of SRLV at physical conditions such as temperatures and given the nature of the virus (i.e. difficult to grow in cell culture) data generated appears of great value. However, we suggest submission as short communication.
INTRODUCTION
The title of the study appears too generic, it could be useful to provide a title that reflects more the study carried out.
- line 78-81: The environmental conditions tested are limited, no humidity or extreme temperatures were considered. We suggest the authors acknowledge in the discussion the limitations of the study, however considering the difficulties in carrying out resistance studies with SRLV
-Please explain which criteria were used to select testing temperatures and time points. Could it be possible that milk is stored at additional temperatures?
MATERIAL AND METHODS
-How The maintenance of temperatures during the experiment was monitored?
-Paragraph 2.5: how many tubes were exposed for each time point?
-Lines 137-140: it is not clear why the Hemacolor®, Merck, Germany was used only on the dried virus.
RESULTS
It would be helpful to have different colors for different lines in all figures as it is a bit hard to follow them.
DISCUSSION
It could be useful to stress the difficulties in titrating SRLV and therefore explaining the selection of only two prototype viruses.
MINOR COMMENTS
ABSTRACT
Line 22: it should be rephrased
INTRODUCTION
Line 33: virus family should be written in italics
- line 37: SRLV genotypes are not four. They are five as stated in the same sentence (A, B, C, D, E).
- line 39-43: It would be better to cite more than one source for this topic.
-line 47: bibliographic reference number 17 was not found in the literature
- line 50: change kits in kids.
- line 64-69: citation needed.
-line 77: citation on SRLV isolation from cell-free milk is needed.
MATERIAL AND METHODS
- line 84-85: the sentence needs to be rewritten; the main verb is missing.
- paragraph 2.5: it is not clear how the dilutions were prepared, it may be useful to provide more detailed information.
-line 109: citation describing MVV cultivation and titration using LSM cells is needed.
RESULTS
- paragraph Abbreviations: the acronym “ns”, which is present in Figures 1a-1b and 2a-2b, is not reported.
-Please delete fig 1a and b embedded in figures 1 and 2.
Author Response
Reviewer #2:
Comments and Suggestions for Authors
MAJOR COMMENTS
The article examines the persistence of small ruminant lentiviruses (SRLVs), in particular Visna Virus (MVV) and caprine encephalitis virus (CAEV). The researchers analysed the duration of infectivity of these viruses under different conditions and found that they persisted longer in milk suspensions than in cell culture media. Even after drying, the viruses retained their ability to replicate for weeks. This suggests that indirect transmission via contaminated fomites or pastures could occur without direct contact between animals. The study covers a gap in data regarding the survival of SRLV at physical conditions such as temperatures and given the nature of the virus (i.e. difficult to grow in cell culture) data generated appears of great value. However, we suggest submission as short communication.
Response 1: We greatly appreciated the thorough work performed by this reviewer, who significantly contributed to improving our manuscript.
We respectfully disagree with the suggestion of downgrading the manuscript to a short communication. As mentioned by the reviewer, this work permits us to close a gap in our knowledge of these viruses, was very demanding in terms of laboratory work, and the results may have a practical impact in formulating biosafety measures for small ruminant flocks.
INTRODUCTION
The title of the study appears too generic, it could be useful to provide a title that reflects more the study carried out.
Response 2: This criticism triggered a vivid discussion, but in the end, all authors agreed with this proposal, and we modified the title as follows:
Testing the tenacity of small ruminant lentiviruses in vitro to assess the potential risk of indirect fomites' transmission.
- line 78-81: The environmental conditions tested are limited, no humidity or extreme temperatures were considered. We suggest the authors acknowledge in the discussion the limitations of the study, however considering the difficulties in carrying out resistance studies with SRLV
-Please explain which criteria were used to select testing temperatures and time points. Could it be possible that milk is stored at additional temperatures?
Response 3: These criticisms are undoubtedly justified, but our work aimed to define the tenacity of these viruses in non-extreme conditions that closely resemble those found in a barn or on a pasture. In the context of potential zoonotic virus transmission, maintaining replicative capacity at more extreme temperatures and humidity conditions, up to and including pasteurization, would have been crucial. Still, for this initial work, we decided to forgo it.
MATERIAL AND METHODS
-How The maintenance of temperatures during the experiment was monitored?
Response 4: This is an important point, and we added the following sentence in lines 123-125:
In our Institute's accreditation frame, temperatures in the laboratories and refrigerators where the viruses were stored were monitored daily.
-Paragraph 2.5: how many tubes were exposed for each time point?
-Lines 137-140: it is not clear why the Hemacolor®, Merck, Germany was used only on the dried virus.
Response 5: The reviewer is correct in asking for these clarifications, and the concerned paragraph was completely modified:
2.5. Virus stored in liquid
Nutrient medium and milk were supplemented with 2.5% Penicillin-Streptomycin (Seraglob, Switzerland, lot 610/134072) and 1% of Amphotericin B (Seraglob, Switzerland, lot 610/763549) to prevent possible bacterial or fungal contamination. A working dilution of viruses was prepared by resuspending viral stocks in a nutrient medium or milk. The dilutions were prepared in a volume of 3 ml for every condition and every foreseeable timepoint and then aliquoted (1ml into 1.5 ml Eppendorf tubes) and stored in the following conditions:
Room temperature (RT) – tubes were stored at ambient room temperature (18 – 25 °C) and were shielded from direct sunlight. 4°C – tubes were stored in a refrigerator in the dark at a stable temperature of 4°C. At the control time points (7, 14, and 21 days), three tubes for each condition were taken and titrated by limiting endpoint dilution. In our Institute's accreditation frame, temperatures in the laboratories and refrigerators where the viruses were stored were monitored daily. To assess the tenacity of the viruses 3 tubes for each both milk and nutrient medium and each temperature were taken and titrated via 10-fold dilution in quadruplicates on 96-well tissue culture plates that were previously seeded with primary cells (LSM for MVV and GSM for CAEV) at 50% confluency.
Plates were incubated for 12 days and stained using the Hemacolor® staining kit (Hemacolor®, Merck, Germany) according to the manufacturer's instructions. At this point, the plates were scored under a brightfield microscope for the presence of syncytia. The Spearman-Kärber method was used to calculate the titers [38].
RESULTS
It would be helpful to have different colors for different lines in all figures as it is a bit hard to follow them.
Response 6: The figures were modified as requested.
DISCUSSION
It could be useful to stress the difficulties in titrating SRLV and therefore explaining the selection of only two prototype viruses.
Response 7: We agree and introduced this sentence in the new discussion:
For our experiments, we chose two prototypic viruses for the two main genotypes, A (MVV 1514) and B (CAEV-CO), aware that this may limit the generalization of our results. Primary isolates would have been ideal, but these viruses often have a strict tropism for macrophages, making their titration extremely cumbersome.
MINOR COMMENTS
ABSTRACT
Line 22: it should be rephrased
Response 8: we agree with the reviewer and modified this sentence as follows (lines 23-24):
These viruses, subjected to harsh treatments such as drying and resuspending, partially maintained their replication capacity.
INTRODUCTION
Line 33: virus family should be written in italics
Response 9: modified accordingly
- line 37: SRLV genotypes are not four. They are five as stated in the same sentence (A, B, C, D, E).
Response 10: modified accordingly
- line 39-43: It would be better to cite more than one source for this topic.
Response 11: An additional reference pointing to the importance of trading in the spread of SRLV was added:
Carrozza, M.L.; Niewiadomska, A.M.; Mazzei, M.; Abi-Said, M.R.; Hue, S.; Hughes, J.; Gatseva, A.; Gifford, R.J. Emergence and pandemic spread of small ruminant lentiviruses. Virus Evol 2023, 9, vead005.
-line 47: bibliographic reference number 17 was not found in the literature
Response 12: We thank the reviewer for spotting this mistake hiding in our EndNote databank. The linked PDF in the databank was correct, but the reference was corrupted. We corrected this reference:
17. Palsson, P.A. Maedi-visna. J Clin Pathol Suppl (R Coll Pathol) 1972, 6, 115–120.
- line 50: change kits in kids.
Response 13: we corrected this spelling mistake.
- line 64-69: citation needed.
Response 14: We added the following reference: Villoria, M.; Leginagoikoa, I.; Luján, L.; Pérez, M.; Salazar, E.; Berriatua, E.; Juste, R.A.; Minguijón, E. Detection of small ruminant lentivirus in environmental samples of air and water. Small Ruminant Res 2013, 110, 155-160.
-line 77: citation on SRLV isolation from cell-free milk is needed.
Response 15: We agree with the reviewer and added the following reference, which we think is the first to mention the isolation of SRLV from cell-free milk.
Adams, D.S.; Klevjer-Anderson, P.; Carlson, J.L.; McGuire, T.C. Transmission and control of caprine arthritis-encephalitis virus. Am. J. Vet. Res 1983, 44, 1670-1675.
MATERIAL AND METHODS
- line 84-85: the sentence needs to be rewritten; the main verb is missing.
Response 16: We agree and modified the sentence: Caprine arthritis encephalitis virus (CAEV-CO), accession number M33677 (CEAVCG), was selected for our experiments.
- paragraph 2.5: it is not clear how the dilutions were prepared, it may be useful to provide more detailed information.
Response 17: We agree and modified the entire paragraph as mentioned in Response 5
-line 109: citation describing MVV cultivation and titration using LSM cells is needed.
Response 18: We thank the reviewer for this suggestion. Surprisingly, we realized this was the first time we had used LSM cells to titrate viruses in a peer-reviewed publication. We regularly use these cells to grow and titrate SRLV-A isolated from sheep and goats but never published these results. What we have already published is the use of these cells to monitor viral replication using RT-PCR to quantify the genome in the supernatant of infected cells or the PERT assay to quantify the RT activity in the supernatants (L. Cardinaux et al. Veterinary Microbiology 162 (2013) 572–581; L. Blatti-Cardinaux et al. / Virology 487 (2016) 50–58). Unfortunately, these techniques do not permit us to measure the infectivity of these supernatants, and they could not be used for this work.
RESULTS
- paragraph Abbreviations: the acronym "ns", which is present in Figures 1a-1b and 2a-2b, is not reported.
Response 19: We added this in the figure legend.
-Please delete fig 1a and b embedded in figures 1 and 2.
Response 20: The figures were modified accordingly.

Reviewer 3 Report
Comments and Suggestions for Authors
This is the review of the manuscript entitled “On the Tenacity of small ruminant lentiviruses” by Samoilenko et al. This manuscript was well written, demonstrating the survival of two SRLV in-vitro under studied conditions.
Minor concerns:
- Line 86- please spell out the GSM cell line as it appears for the first time in the text. It can then be removed from line 106.
- Line 88- should be part of the paragraph above.
- Line 93- should be part of the paragraph above.
- Line 116-120- they should be written in sentences as part of the paragraph above.
- Line 135- Can you verify that the 1 ml media was replaced with only 100ul of fresh media. The volume seems to be too low for a 12-days incubation?
- The manuscript ended abruptly. It needs a conclusion sentence or paragraph.
Author Response
Reviewer #3:
Comments and Suggestions for Authors
This is the review of the manuscript entitled “On the Tenacity of small ruminant lentiviruses” by Samoilenko et al. This manuscript was well written, demonstrating the survival of two SRLV in-vitro under studied conditions.
Response 1: We thank the reviewer for this effort, which was very helpful in improving our manuscript.
Minor concerns:
1. Line 86- please spell out the GSM cell line as it appears for the first time in the text. It can then be removed from line 106.
Response 2: The reviewer is correct; we have changed the manuscript accordingly (lines 89 and 109).
2. Line 88- should be part of the paragraph above.
Response 3: We agree and changed the manuscript accordingly
3. Line 93- should be part of the paragraph above.
Response 4: We agree and changed the manuscript accordingly
4. Line 116-120- they should be written in sentences as part of the paragraph above.
Response 5: We agree and changed the manuscript accordingly
5. Line 135- Can you verify that the 1 ml media was replaced with only 100ul of fresh media. The volume seems to be too low for a 12-days incubation?
Response 6: We agree entirely and modified the manuscript as follows (lines145-149):
The dried residue in each well was resuspended by adding 1ml of DMEM+FBS, and 100 ul of the resuspended material was transferred into an Eppendorf tube containing 900 ul of fresh cell culture medium. A 10-fold dilution of the material was performed and finally transferred in quadruplicates to 96-well tissue culture plates seeded with primary cells (LSM for MVV and GSM for CAEV) at 50% confluency.
6. The manuscript ended abruptly. It needs a conclusion sentence or paragraph.
Response 7: We agree with our reviewer and attempted to smooth the end of our manuscript as follows:
In conclusion, we must emphasize that our results do not yet allow us to determine the actual risk of SRLV transmission via fomites. Indeed, to answer this question, it will be necessary to establish the minimal infectious titer of these viruses in vivo, especially by the oral route. Indeed, the required infectious titers to infect an animal were shown to be very low for intratracheal infection but estimated to be much higher for intranasal infection and the oral route [29,42].
